# THE UNCERTAINTY-PERCEPTION TRADEOFF

## ABSTRACT

Generative models have achieved groundbreaking performance in restoration tasks and inverse problems, producing results that are often indistinguishable from real data. Yet these models are also known to produce hallucinations, or artifacts that are not present in the original input, raising concerns about the uncertainty of the models' predictions. In this paper we study this phenomenon, employing information-theory tools to reveal a fundamental tradeoff between perception and uncertainty. Our mathematical analysis shows that as perceptual quality increases, so does the uncertainty of a restoration algorithm as quantified by error entropy. We derive and illustrate the behavior of the uncertainty-perception function, showcasing both local and global bounds that define the the feasible region of the tradeoff. Furthermore, we revisit a well-known relation between estimation distortion and uncertainty and generalize its scope to include perception quality, thereby shedding new light on the well-established perception-distortion tradeoff. Our work offers a principled analysis of uncertainty, highlighting its interplay with perception and the limitations of generative models in restoration tasks.

## 1 INTRODUCTION

Generative artificial intelligence (AI) has revolutionized the field inverse problems in recent years. Deep learning models have achieved unprecedented performance in restoration tasks, producing results that are often indistinguishable from real data across various tasks such as image denoising, super-resolution, and inpainting, pushing the boundaries of what was previously attainable. Their ability to infer missing information and restore corrupted data has far-reaching implications in a variety of fields, including medical imaging, computer vision, and signal processing.

While powerful, generative models are susceptible to a phenomenon known as *"hallucinations"*, characterized by the generation of highly detailed realistic content that appears authentic but deviates from the original input data, hindering applications where faithfulness is crucial. The underlying cause of hallucination lies in the ill-posed nature of restoration problems, where multiple possible solutions can explain the observed measurements, leading to uncertainty in the estimation process. Interestingly, the severity of hallucination appears to be correlated with the perceptual quality of the generative model. Despite this observation, a rigorous theoretical framework linking perception (Blau & Michaeli, 2018) and uncertainty remains elusive. This raises a fundamental question:

*Can one design an AI model that achieves both high perceptual quality and low uncertainty?*

In this paper, we address the question above and demonstrate that the two objectives are inherently at odds with each other, establishing a tradeoff between uncertainty and perception. Our main contribution are as follows: (i) We introduce the uncertainty-perception (UP) function, grounded in information-theoretic principles, to prove the existence of an inherent tradeoff between uncertainty and perception that holds true for any underlying data distribution, inverse problem or restoration model (Theorem 1). (ii) Employing Rényi divergence as a measure of perception, we derive properties the UP function, giving rise to the uncertainty-perception plane that categorizes algorithms into three distinct performance domains (Theorem 3). (iii) We establish a relationship between uncertainty and distortion, showing that the uncertainty-perception tradeoff implies the well-known distortion-perception tradeoff (Theorem 4). To support our theoretical findings, we provide a numerical illustration of the tradeoff in established algorithms for single image super-resolution. Thus, developers should acknowledge this tradeoff when designing restoration algorithms in practice, prioritizing either high perceptual quality or low uncertainty to align with specific requirements.

## 2 RELATED WORK

**Uncertainty Quantification** Uncertainty quantification techniques can be broadly categorized into two main paradigms: Bayesian estimation and frequentist approaches. The Bayesian paradigm defines uncertainty by assuming a distribution over the model parameters and/or activation functions. The most prevalent approach is Bayesian neural networks (MacKay, 1992; Valentin Jospin et al., 2020; Izmailov et al., 2020), which are stochastic models trained using Bayesian inference. To improve efficiency, approximation methods have been developed, including Monte Carlo dropout (Gal & Ghahramani, 2016; Gal et al., 2017a), stochastic gradient Markov chain Monte Carlo (Salimans et al., 2015; Chen et al., 2014), Laplacian approximations (Ritter et al., 2018) and variational inference (Blundell et al., 2015; Louizos & Welling, 2017; Posch et al., 2019). Alternative Bayesian techniques encompass deep Gaussian processes (Damianou & Lawrence, 2013), deep ensembles (Ashukha et al., 2020; Hu et al., 2019), and deep Bayesian active learning (Gal et al., 2017b). Abdar et al. (2021) provides an extensive review of Bayesian uncertainty quantification.

In contrast to Bayesian methods, frequentist approaches operate under the assumption of fixed model parameters with no underlying distribution. Examples of such distribution-free techniques are model ensembles (Lakshminarayanan et al., 2017; Pearce et al., 2018), bootstrap (Kim et al., 2020; Alaa & Van Der Schaar, 2020), interval regression (Pearce et al., 2018; Kivaranovic et al., 2020; Wu et al., 2021) and quantile regression (Gasthaus et al., 2019; Romano et al., 2019).

An emerging approach in recent years is conformal prediction (Angelopoulos & Bates, 2021; Shafer & Vovk, 2008), which leverages a labeled calibration dataset to convert point estimates into prediction regions. Conformal methods are versatile, require no retraining, are computationally efficient, and provide coverage guarantees in finite samples (Lei et al., 2018). These works include conformalized quantile regression (Romano et al., 2019; Sesia & Candès, 2020; Angelopoulos et al., 2022b), conformal risk control (Angelopoulos et al., 2022a; Bates et al., 2021; Angelopoulos et al., 2021), and semantic uncertainty intervals for generative adversarial networks (Sankaranarayanan et al., 2022). Kutiel et al. (2023) introduces the notion of conformal prediction masks, interpretable image masks with rigorous statistical guarantees for image restoration, highlighting regions of high uncertainty in the recovered images. Please see (Sun, 2022) for an extensive survey of distribution-free conformal prediction methods. A recent approach (Belhasin et al., 2023) introduces a principal uncertainty quantification method for image restoration that considers spatial relationships within the image to derive uncertainty intervals that are guaranteed to include the true unseen image with a user-defined confidence probabilities. While the above studies offer a variety of approaches for quantifying uncertainty, a rigours analysis of the relationship between uncertainty and perception remains underexplored, particularly in the context of inverse problems.

**Perception Quantification** Perceptual quality in restoration tasks encompasses how humans perceive the output, considering visual fidelity, similarity to the original, and absence of artifacts. While traditional metrics like PSNR and SSIM (Wang et al., 2004) capture basic similarity, they miss finer details and higher-level structures. Learned metrics like LPIPS (Zhang et al., 2018), VGG-loss (Simonyan & Zisserman, 2014), and DISTS (Ding et al., 2020) offer improvements but still operate on pixel or patch level, potentially overlooking holistic aspects. Recently, researchers have leveraged image-level embeddings from large vision models like DINO (Caron et al., 2021) and CLIP (Radford et al., 2021) to capture high-level similarity. In this study, we adopt a mathematical definition of perceptual quality based on the divergence between probability density functions, as proposed by Blau & Michaeli (2018) and further explored by Hepburn et al. (2021).

**The Distortion-Perception Tradeoff** The most relevant study to our research is the work on the distortion-uncertainty tradeoff (Blau & Michaeli, 2018) and its follow-ups (Freirich et al., 2021; Blau & Michaeli, 2019; Blau et al., 2018). Blau & Michaeli (2018) established a convex tradeoff between perceptual quality and distortion in image restoration, applicable to any distortion measure and distribution. When using Mean Squared Error (MSE), they found that perfect perceptual quality can be obtained at the expense of no more than 3dB in peak signal-to-noise ratio. Freirich et al. (2021) extends this, providing closed-form expressions for the tradeoff when MSE distortion and the Wasserstein-2 perception distance are considered. In this setting, optimal estimators can be obtained by simply interpolating between the results of minimum MSE estimators and perfect-perceptual estimators. Thus, our works complements these studies by analyzing the relation of uncertainty to perception and distortion and providing a new perspective on the distortion-perception tradeoff.

## 3 PRELIMINARIES

To make the paper self-contained, we provide a brief overview of essential definitions and fundamental results that stand in the center of our study. Let $X$, $Y$ and $Z$ be continuous random variables with probability density functions $p_X(x)$, $p_Y(y)$ and $p_Z(z)$ respectively, defined over a space $\Omega$.

**Definition 1** (**Entropy**). *The differential entropy of $X$, whose support is a set $S_x$, is defined by*

$$h(X) \triangleq - \int_{S_X} p_X(x) \log p_X(x) dx.$$

**Definition 2** (**Rényi Entropy**). *The Rényi entropy of order $r \geq 0$ of $X$ is defined by*

$$h_r(X) \triangleq \frac{1}{1-r} \ln \int p_X^r(x) dx.$$

*The above quantity generalizes various notions of entropy, including Hartley entropy, collision entropy, and min-entropy. In particular, for $r = 1$ we have*

$$h_1(X) \triangleq \lim_{r \to 1} h_r(X) = h(X).$$

**Definition 3** (**Entropy Power**). *Let be $h(X)$ be the differential entropy of $X \in \mathbb{R}^d$. Then, the entropy Power of $X$ is given by*

$$N(X) \triangleq \frac{1}{2\pi e} e^{\frac{2}{d} h(X)}.$$

**Definition 4** (**Divergence**). *A statistical divergence is any function $D_v : \Omega \times \Omega \to \mathbb{R}^+$ which satisfies the following conditions for all $p, q \in \Omega$:*

    *1. $D_v(p, q) \geq 0$.    2. $D_v(p, q) = 0$ iff $p = q$ almost everywhere.*

**Definition 5** (**Rényi Divergence**). *The Rényi divergence of order $r \geq 0$ between $p_X$ and $p_Y$ is*

$$D_r(X, Y) \triangleq \frac{1}{r-1} \ln \int p_X^r(x) p_Y^{1-r}(x) dx.$$

*The above establishes a spectrum of divergence measures, generalising the Kullback–Leibler divergence as $D_1(X, Y) = D_{KL}(X, Y)$.*

**Definition 6** (**Conditioning**). *Consider the joint probability $p_{XY}$ and the conditional probabilities $p_{X|Y}(x|y)$ and $p_{Z|Y}(z|y)$. The conditional differential entropy of $X \in \mathbb{R}^d$ given $Y$ is defined as*

$$h(X|Y) \triangleq - \int_{S_{XY}} p_{XY}(x, y) \log p_{X|Y}(x|y) dx dy = \mathbb{E}_{y \sim p_Y} \left[ h(X|Y = y) \right]$$

*where $S_{XY}$ is the support set of $p_{XY}$. Then, the conditional entropy power of $X$ given $Y$ is*

$$N(X|Y) = \frac{1}{2\pi e} e^{\frac{2}{d} h(X|Y)}.$$

*Similarly, the conditional divergence between $X$ and $Z$ given $Y$ is defined as*

$$D_v(X, Z|Y) \triangleq \mathbb{E}_{y \sim p_Y} \left[ D_v(X|Y = y, Z|Y = y) \right].$$

*For example, the conditional Rényi divergence is given by*

$$D_r(X, Z|Y) \triangleq \int \left( \frac{1}{r-1} \ln \int p_{X|Y}^r(x|y) p_{Z|Y}^{1-r}(x|y) dx \right) p_Y dy.$$

For the remainder of this paper, we assume that the aforementioned quantities, which involve integrals, are well-defined and finite. In Table 1, we provide closed-form expressions for a number of the above quantities associated with the multivariate Gaussian distribution. Below, we present two important results that are used throughout our derivations.

**Lemma 1** (**Maximum Entropy Principle** (Cover, 1999)). *Let $X \in \mathbb{R}^d$ be a continuous random variable with zero mean and covariance $\Sigma_x$. Define $X_G \sim \mathcal{N}(0, \Sigma_x)$ to be a Gaussian random variable, independent of $X$, with the identical covariance matrix $\Sigma_{x_G} = \Sigma_x$. Then,*

$$h(X) \leq h(X_G),$$

$$N(X) \leq N(X_G) = |\Sigma_x|^{1/d}.$$

Table 1: Formulas for Multivariate Gaussian Distribution

| Distribution | Quantity | Closed-Form Expression |
|---|---|---|
| $X \sim \mathcal{N}(\mu_x, \Sigma_x)$ | $h(X)$ | $\frac{1}{2}\ln\{(2\pi e)^d |\Sigma_x|\}$. |
| $X \sim \mathcal{N}(\mu_x, \Sigma_x)$ | $N(X)$ | $|\Sigma_x|^{1/n}$. |
| $X \sim \mathcal{N}(\mu_x, \Sigma_x)$ | $h_{\frac{1}{2}}(X)$ | $\frac{1}{2}\ln\{(8\pi)^d |\Sigma_x|\}$. |
| $X \sim \mathcal{N}(\mu_x, \Sigma_x)$, $Y \sim \mathcal{N}(\mu_y, \Sigma_y)$ | $D_{1/2}(X, Y)$ | $\frac{1}{4}(\mu_x - \mu_y)^T \left(\frac{\Sigma_x + \Sigma_y}{2}\right)^{-1}(\mu_x - \mu_y) + \ln\left(\frac{\left|\frac{\Sigma_x + \Sigma_y}{2}\right|}{\sqrt{|\Sigma_x||\Sigma_y|}}\right)$. |

**Lemma 2** (**Entropy power inequality** (Madiman et al., 2017)). *Let $X$ and $Y$ be independent continuous random variables. Then, the following inequality holds*

$$N(X) + N(Y) \le N(X + Y),$$

*where equality holds iff $X$ and $Y$ are multivariate Gaussian random variables with proportional covariance matrices. Equivalently, let $X_g$ and $Y_g$ be defined as independent, isotropic multivariate Gaussian random variables satisfying $h(X_g) = h(X)$ and $h(Y_g) = h(Y)$. Then,*

$$h(X) + h(Y) = h(X_g) + h(Y_g) = h(X_g + Y_g) \le h(X + Y).$$

## 4 THE UNCERTAINTY-PERCEPTION TRADEOFF

### 4.1 PROBLEM FORMULATION

We consider the problem of recovering a random vector $X \in \mathbb{R}^d$ from its observations, represented by another random vector $Y = M(X) \in \mathbb{R}^{d'}$ where $M : \mathbb{R}^d \to \mathbb{R}^{d'}$ is a non-invertible measurement function. This problem translates to building an estimator $\hat{X}(Y)$ which induces a conditional distribution measure $p_{\hat{X}|Y}$ on $\mathbb{R}^d$. We rely on the following mild assumptions:

**Assumption 1** (Loss of Information). *The problem is ill-posed so $X$ cannot be perfectly recovered from $Y$. Namely, $p_{X|Y}(\cdot|y)$ is not a delta function for almost every $y$.*

**Assumption 2** (Markovian Process). *The estimation process is a Markov chain $X \to Y \to \hat{X}$, such that $\hat{X}$ is independent of $X$ given $Y$.*

**Assumption 3** (Unbiasedness). *$\hat{X}$ is an unbiased estimator of $X$, implying $\mathbb{E}(\hat{X}) = \mathbb{E}(X)$.*

Assumptions 1 and 2 are standard in the field of inverse problems (Blau & Michaeli, 2018; Freirich et al., 2021). The first arises from the non-invertibility of the observation process, while the second implies the estimator efficiently extracts all relevant information about $X$ from $Y$, as access to the true signal is unavailable. Assumption 3 is made without loss of generality as the chosen uncertainty measure, defined later, exhibits translation-invariance and thus is unaffected by bias.

We are interested in studying estimators $\hat{X}(Y)$ with respect to two performance criteria: uncertainty and perception. For perceptual quality, we adopt a similar approach to the mathematical definition proposed by Blau & Michaeli (2018) and measure perception by a conditional divergence $D_v(X, \hat{X}|Y)$. Using the abstract measure $U_{nc}(\hat{X}|Y)$ to represent the uncertainty of an estimator $\hat{X}$ given the information in $Y$, we formulate the following uncertainty-perception (UP) function:

$$U(P) \triangleq \min_{p_{\hat{X}|Y}} \left\{ U_{nc}(\hat{X}|Y) \ : \ D_v(X, \hat{X}|Y) \le P \right\}. \tag{1}$$

In words, $U(P)$ is the minimal uncertainty that can be attained by an estimator with perception quality of at least $P$, given the information in the measurements $Y$.

There exist diverse approaches to define and quantify uncertainty (Gawlikowski et al., 2023; Abdar et al., 2021). In this paper, we focus on a fundamental information-theoretic approach, relying on

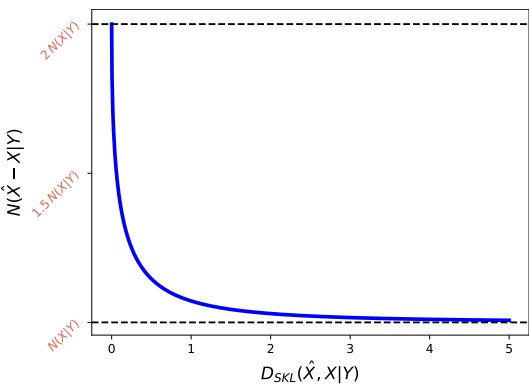

Figure 1: The uncertainty-perception tradeoff for the setting of example 1.

the concept of entropy that measures the statistical dispersion of a random variable. Intuitively, an uncertainty quantity should be non-negative $U_{nc}(\hat{X}|Y) \geq 0$, yet, the differential entropy may be negative. Therefore, entropy power offers a more natural measure of uncertainty, leading to:

$$U(P) \triangleq \min_{p_{\hat{X}|Y}} \left\{ N(\hat{X} - X|Y) \ : \ D_v(X, \hat{X}|Y) \leq P \right\}. \tag{2}$$

The chosen objective extracts the information content of the error signals rather than simply their energy (second order statistics), compared with distortion measures. Furthermore, for a fixed perception index $P$, the minimization above promotes concentrated errors, yielding robust predictions.

We remark that an alternative formulation may be

$$\tilde{U}(P) \triangleq \min_{p_{\hat{X}|Y}} \left\{ N(\hat{X} - X) \ : \ D_v(X, \hat{X}) \leq P \right\}. \tag{3}$$

However, we highlight that the objective function satisfies $N(\hat{X} - X|Y) \leq N(\hat{X} - X)$ where equality holds if and only if the error $\mathcal{E} = \hat{X} - X$ is independent of $Y$. Thus, it may overestimate the uncertainty in the error. While further investigation is warranted, we hypothesize that the behavior of function (3) mirrors that of the tradeoff function (2), which we examine in the following section.

### 4.2 The Uncertainty-Perception Plane

Thus far, we have formulated the uncertainty-perception function and explained its underlying rationale. We now proceed to derive its fundamental properties to establish the uncertainty-perception tradeoff. To demonstrate the nature of this function, we begin with an illustrative example.

**Example 1.** *Consider $Y = X + W$ where $X \sim \mathcal{N}(0, 1)$ and $W \sim \mathcal{N}(0, \sigma^2)$ are independent. Let the perception measure be the symmetric KL divergence and assume stochastic estimators of the form $\hat{X} = \mathbb{E}[X|Y] + Z$ where $Z \sim \mathcal{N}(0, \sigma_z^2)$ is independent of $Y$. As derived in Appendix A, the uncertainty-perception function admits a closed form expression in this case, given by*

$$U(P) = N(X|Y) \left[ 1 + \left( P + 1 - \sqrt{(P+1)^2 - 1} \right)^2 \right], \text{ where } N(X|Y) = \sigma^2/(1 + \sigma^2).$$

The above example suggests a structure for uncertainty-perception function $U(P)$, which inherently relies on $N(X|Y)$. As shown in Figure 1, the minimal attainable uncertainty increases as the perception quality improves, clearly demonstrating the tradeoff. Yet, the precise form of the tradeoff is dictated in general by the underlying distributions of $X$ and $Y$, along with the specific divergence measure $D_v(\cdot, \cdot)$ employed. Fortunately, the following theorem establishes overarching properties of the uncertainty-perception function, $U(P)$, that hold irrespective of these specific distributions and divergence measures.

**Theorem 1.** *The uncertainty-perception function $U(P)$ displays the following properties*

1. *Quasi-linearity (monotonically non-increasing and continuous):*

$$\min \left( U(P_1), U(P_2) \right) \leq U \left( \lambda P_1 + (1 - \lambda) P_2 \right) \leq \max \left( U(P_1), U(P_2) \right). \tag{4}$$

*2. Boundlessness:*

$$N(X|Y) \leq U(P) \leq 2N(X_G|Y). \tag{5}$$

*where $X_G$ is as defined in Lemma (1). Here $N(X|Y)$ represents the inherent uncertainty of the problem and $N(X_G|Y)$ is its upper bound that depends on the deviation of $X$ from Gaussianity.*

The aforementioned theorem establishes the existence of a tradeoff between perceptual quality and uncertainty for any divergence measure, underlying data distributions, inverse problem or restoration model. The tradeoff is fundamentally related to $N(X|Y)$, uncertainty arising from the information loss during the observation process (Assumption 1). The upper bound depends on $N(X|Y)$ as it can be expressed as

$$N(X_G|Y) = N(X|Y)e^{\frac{2}{d}D_{KL}(X,X_G|Y)}. \tag{6}$$

This equation shows that as $X$ approaches Gaussianity, $N(X|Y)$ approaches $N(X_G|Y)$. However, concurrently, it implies in general higher values of $N(X|Y)$ due to Lemma 1. The significance of this finding lies in the surprising fact that for multivariate Gaussian distributions, perfect perceptual quality can be attained at the cost of twice the inherent uncertainty of the problem. This extends to any distribution of $X$ when $d$ is sufficiently large so $e^{\frac{2}{d}D_{KL}(X,X_G|Y)} \approx 1$.

While Theorem 1 outlines important characteristics of the uncertainty-perception function, further insights require additional assumptions. The theorem below addresses the optimization process for a fixed perceptual index $P$.

**Theorem 2.** *Assume $D_v(X, \hat{X}|Y)$ is convex in its second argument. Then, for any $P \geq 0$, the minimum is attained on the boundary where $D_v(X, \hat{X}|Y) = P$.*

The above result is promising as it suggests that the optimization process can be confined to the constraint set's boundary, facilitating the optimization task. We continue by adopting a specific divergence function. Given that our minimization objective involves entropy, Rényi divergence emerges as a natural choice for our perception measure. Specifically, for $r = 1/2$, we arrive at:

$$U(P) = \min_{p_{\hat{X}|Y}} \left\{ N(\hat{X} - X|Y) \ : \ D_{1/2}(X, \hat{X}|Y) \leq P \right\}. \tag{7}$$

Rényi divergence plays a critical role in the proofs of Bayesian estimators and numerous information theory calculations (Van Erven & Harremos, 2014). It is directly related to Rényi entropy, which generalizes various notions of entropy, including Hartley entropy, Shannon entropy, collision entropy, and min-entropy. Moreover, while we set $r = 1/2$ to facilitate our derivations, it is important to note that all orders $r \in (0, 1)$ are equivalent (Van Erven & Harremos, 2014), since

$$\frac{r}{t}\frac{1-t}{1-r}D_t(\cdot, \cdot) \leq D_r(\cdot, \cdot) \leq D_t(\cdot, \cdot), \ \forall 0 < r \leq t < 1. \tag{8}$$

Consequently, analyzing the specific formulation provided by (7) may yield valuable insights applicable to a wide range of divergence measures. The next theorem provide bounds for the tradeoff.

**Theorem 3.** *The uncertainty-perception function is confined to the following region*

$$\eta(P) \cdot N(X|Y) \ \leq \ U(P) \ \leq \ \eta(P) \cdot N(X_G|Y)$$

*where $1 \leq \eta(P) \leq 2$ is a convex function w.r.t the perception index and is given by*

$$\eta(P) = \left( 2e^{\frac{2P}{d}} - \sqrt{(2e^{\frac{2P}{d}} - 1)^2 - 1} \right).$$

It is noteworthy that Theorem 3 holds true regardless of the underlying distributions of $X$ and $Y$, thereby providing a universal characterization of the tradeoff as a function of perception. Furthermore, as depicted in Figure 2, Theorem 3 gives rise to the uncertainty-perception plane, which divides the space into three distinct regions:

1. Impossible region, where no estimator can reach.
2. Optimal region, encompassing all estimators that are optimal according to (7).
3. Suboptimal region of estimators which exhibit overly high uncertainty.

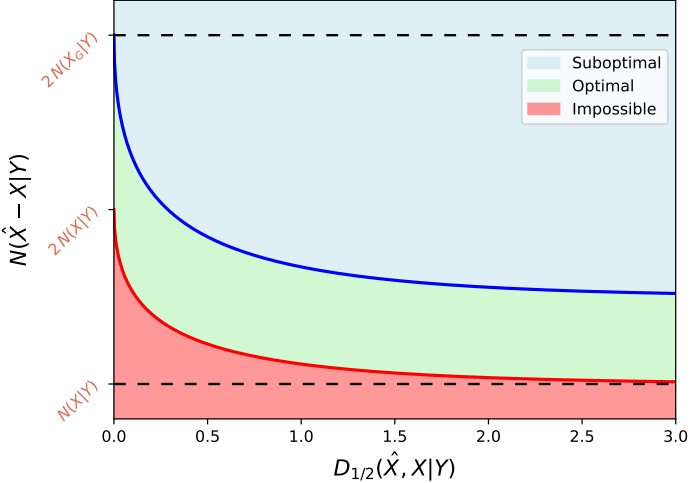

Figure 2: The uncertainty-perception plane.

The existence of an impossible region highlights the uncertainty-perception tradeoff, proving no estimator can achieve both high perception and low uncertainty simultaneously. Furthermore, the uncertainty-perception plane may provide a valuable tool for evaluating estimator performance and identifying opportunities for improvement. Estimators residing in the suboptimal region can potentially be optimized to achieve lower uncertainty without sacrificing perceptual quality. It is important to note that some suboptimal estimators may also exist within the optimal region. In addition, we conjecture that the general form of the tradeoff, given by the inequality in Theorem 3, remains valid for different divergence measures, with the specific form of $\eta(P)$ capturing the nuances of each measure. For instance, if we consider the Hellinger distance as our perception measure, we obtain the same inequality but with $\eta(P)$ defined for $0 \leq P \leq 1$ as

$$\eta_{\text{Hel}}(P) = \frac{2}{(1-P)^{4/d}} - \sqrt{\left(\frac{2}{(1-P)^{4/d}} - 1\right)^2 - 1}. \tag{9}$$

Next, we emphasize the dependency of the tradeoff on dimension of the random variable. To that end, we consider $\eta(P)$ as a function of dimension for a fixed perceptual quality, denoted as $\eta(d; P)$. As shown in Fig. 3, $\eta(d; P)$ exhibits a rapid incline as $d$ increases, and the increment is more pronounced for higher perception qualities. This observation suggests that in high-dimensional settings, the uncertainty-perception tradeoff becomes more severe, implying that any marginal improvement in perception for an algorithm is accompanied by a dramatic increase in uncertainty.

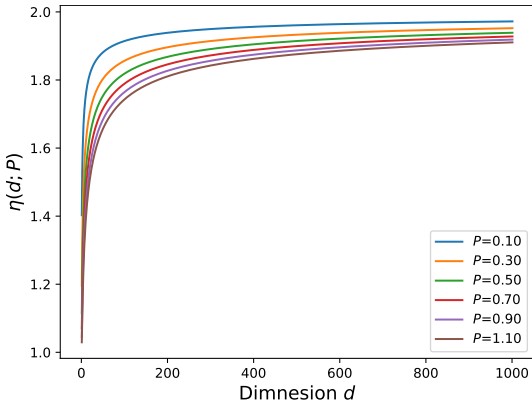

Figure 3: Impact of dimensionality on the uncertainty-perception tradeoff.

Lastly, while the uncertainty-perception tradeoff arises from theory, it has immediate practical implications. Knowing that high perceptual quality and low uncertainty cannot coexist, developers should prioritize one over the other based on specific application requirements.

### 4.3 THE DISTORTION-PERCEPTION TRADEOFF

Having established the uncertainty-perception tradeoff and its characteristics, we now boraden our analysis to estimation distortion, particularly the mean squared-error (MSE). A well-known result estimation theory states that for any random variable $X$ and any estimator $\hat{X}$ given side information $Y$, the following holds true (Cover, 1999):

$$\mathbb{E}\left[||\hat{X} - X||^2\right] \geq \frac{1}{2\pi e}e^{2h(X|Y)}. \tag{10}$$

This inequality, related to the uncertainty principle, serves as a fundamental limit to the minimal MSE achieved by any estimator. However, it does not consider the estimation uncertainty of $\hat{X}$ as the right hand side is independent of $\hat{X}$. Thus, we extend the above in the following theorem.

**Theorem 4.** *For any random variable $X$, observation $Y$ and unbiased estimator $\hat{X}$, it holds that*

$$\frac{1}{d}\mathbb{E}\left[||\hat{X} - X||^2\right] \geq N\left(\hat{X} - X|Y\right).$$

Notice that for any estimator $\hat{X}$, the inequality $N(\hat{X} - X|Y) \geq N(X|Y)$ holds true, implying that

$$\frac{1}{d}\mathbb{E}[||\hat{X} - X||^2|Y] \geq N(X|Y) = \frac{1}{2\pi e}e^{\frac{2}{d}h(X|Y)}. \tag{11}$$

This result aligns with equation (10), demonstrating that Theorem 4 serves as a generalization of inequality (10), incorporating the uncertainty associated with estimation. Furthermore, by considering the estimator $\hat{X}$ as a function of the perception index $P$, we can derive the next corollary.

**Corollary 1.** *Define the distortion-perception function as*

$$D(P) \triangleq \min_{p_{\hat{X}|Y}} \left\{\frac{1}{d}\mathbb{E}\left[||\hat{X} - X||^2\right] : \ D_v(X, \hat{X}|Y) \leq P\right\}.$$

*Then, for any perceptual index $P$, we have $D(P) \geq U(P)$.*

Thus, when utilizing MSE as a measure of distortion, the uncertainty-perception tradeoff induces a distortion-perception tradeoff (Blau & Michaeli, 2018), offering a novel interpretation of the latter. This insight underlines the fundamental connection between uncertainty, distortion, and perception.

## 5 NUMERICAL ILLUSTRATION

We experimentally illustrates the uncertainty-perception tradeoff and its connection to MSE distortion in the context of super-resolution (SR) on the BSD100 dataset (Martin et al., 2001), following previous works (Freirich et al., 2021; Blau & Michaeli, 2018). We compare various SR algorithms, including bicubic, EDSR (Lim et al., 2017), ESRGAN (Wang et al., 2018), SinGAN (Shaham et al., 2019), SANGAN (Kligvasser & Michaeli, 2021), DIP (Ulyanov et al., 2018), SRResNet/SRGAN variants (Ledig et al., 2017), EnhanceNet (Sajjadi et al., 2017), and LDMs with parameter $\beta \in [0, 1]$ (Rombach et al., 2022), where $\beta = 0$ recovers DDIM (Ho et al., 2020) and $\beta = 1$ recovers DDPM (Song et al., 2020).

Estimating high-dimensional statistics poses challenges and is susceptible to error (Laparra et al., 2020). To address this, we treat images as stationary random sources and extract $9 \times 9$ patches for our calculations. We employ kernel density estimation for Rényi divergence and compute it via empirical expectations. Additionally, we use the Kozachenko-Leonenko estimator based on nearest neighbor distances to calculate the patch sample differential entropy (Kozachenko & Leonenko, 1987; Delattre & Fournier, 2017; Beirlant et al., 1997; Marin-Franch & Foster, 2012).

Figure 4 (left) shows the aforementioned SR methods in the uncertainty-perception plane. An unattainable blank region in the lower left corner signifies the tradeoff: no model achieves both

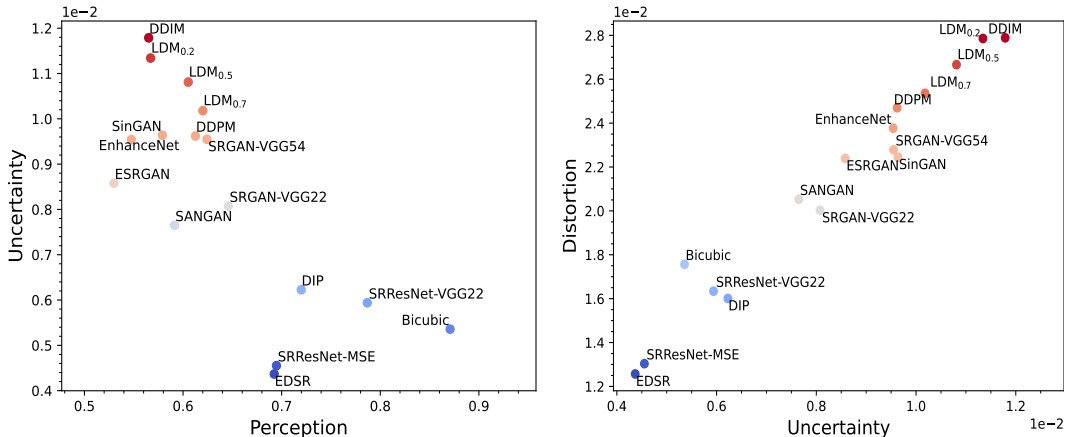

Figure 4: Evaluation of SR algorithms on (left) the uncertainty-distortion plane and (right) on the uncertainty-distortion plane.

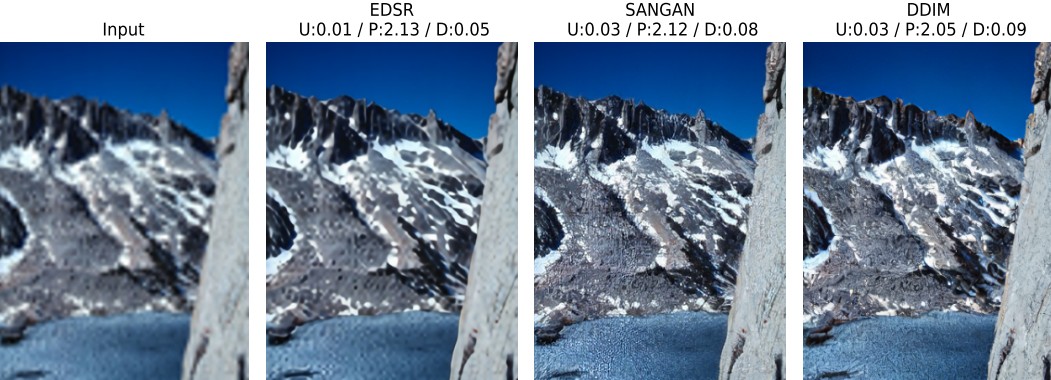

Figure 5: Visual comparison of algorithms on the uncertainty-perception plane, with uncertainty (U), perception (P), and MSE distortion (D) measures shown. Algorithms are ordered from low to high uncertainty (left to right), with an accompanying increase in perceptual quality.

low uncertainty and high perceptual quality. We observe an anti-correlation near this region; modest improvements in perceptual quality lead to dramatic increases in uncertainty, further suggesting a tradeoff that intensifies in high-dimension ($d = 243$). To confirm the relationship between uncertainty and distortion, we plot the same algorithms on the uncertainty-distortion plane. Figure 4 (right) clearly demonstrates on the right that any increase in uncertainty leads to a significant rise in distortion, reinforcing our observations from the previous section. Finally, Figure 5 depicts the outputs of several algorithms lying across the uncertainty-perception plane.

## 6 CONCLUSION

In this study, we formulated and established the uncertainty-perception tradeoff in restoration tasks based on information-theory tools. Namely, achieving high perceptual quality entails high uncertainty levels. We provided a comprehensive characterization of this tradeoff for Rényi divergence, revealing its quasi-linear nature and its pivotal dependence on dimensionality. We presented the uncertainty-perception plane which partitions the space and thus provides an effective tool for assessing estimator performance and identifying areas of improvement. By establishing a direct link between uncertainty and MSE distortion, we have offered a fresh interpretation of the well-known uncertainty-distortion tradeoff. Thus, our work highlights the fundamental interplay between uncertainty, distortion, and perception. Lastly, exploring the role of information rate within the tradeoff presents a promising direction for future research.

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

## A  DERIVATION OF EXAMPLE 1

Since $\hat{X} = \mathbb{E}\left[X|Y\right] + Z$, then $\hat{X}|Y \sim \mathcal{N}(\mathbb{E}\left[X|Y\right], \sigma_z^2)$. Moreover, $X|Y \sim \mathcal{N}(\mathbb{E}\left[X|Y\right], \sigma_q^2)$ where $\sigma_q^2 = \frac{\sigma 2}{1+\sigma^2}$. Thus, the conditional error entropy is given by $N(\hat{X} - X|Y) = \sigma_q^2 + \sigma_z^2$ and the symmetric KL divergence is $D_{SKL}(X, \hat{X}|Y) = \frac{\sigma_q^2 + \sigma_z^2}{2\sigma_z\sigma_q} - 1$, leading the following problem

$$U(P) = \min_{\sigma_z} \left\{ \sigma_q^2 + \sigma_z^2 \; : \; \frac{\sigma_q^2 + \sigma_z^2}{2\sigma_z\sigma_q} - 1 \leq P \right\}. \tag{12}$$

Therefore, we seek the minimal value of $\sigma_z$ that satisfies the constraint. Note that the minimal value is attained at the boundary of the constraint set, where the inequality becomes an equality

$$\frac{\sigma_q^2 + \sigma_z^2}{2\sigma_z\sigma_q} - 1 = P \; \Rightarrow \; \sigma_z^2 - 2\sigma_q(P+1)\sigma_z + \sigma_q^2 = 0. \tag{13}$$

The solution to the aforementioned quadratic problem is $\sigma_z^* = \sigma_q \left( P + 1 - \sqrt{(P+1)^2 - 1} \right)$. Substituting the later into the objective function, we obtain

$$U(P) = \sigma_q^2 \left[ 1 + \left( P + 1 - \sqrt{(P+1)^2 - 1} \right)^2 \right]. \tag{14}$$

Finally, the entropy power of an univariate Gaussian distribution equals its variance $\sigma_q^2 = N(X|Y)$.

## B  PROOF OF THEOREM 1

First, the constraint $\mathcal{C}(P) \triangleq \{ \hat{X} \ : \ D_v(X, \hat{X}|Y) \leq P \}$ defines a compact set which is continuous in $P$. Hence, by the Maximum Theorem (Cover, 1999), $U(P)$ is continuous. In addition, $U(P)$ is the minimal error entropy power obtained over a constraint set whose size does not decrease with $P$, thus, $U(P)$ is non-increasing in $P$. Any continuous non-increasing function is quasi-linear. For the lower bound consider the case where $P = \infty$, leading to the following unconstrained problem

$$U(\infty) \triangleq \min_{p_{\hat{X}|Y}} N(\hat{X} - X|Y). \tag{15}$$

For any $P \geq 0$ it holds that $U(\infty) \leq U(P)$, and by Lemma 2 we have

$$N(X|Y) + \min_{p_{\hat{X}|Y}} N(\hat{X}|Y) \leq U(\infty). \tag{16}$$

Since $\min_{p_{\hat{X}|Y}} N(\hat{X}|Y) \geq 0$ we obtain

$$\forall P \geq 0 : \quad N(X|Y) \leq U(P). \tag{17}$$

Next, we have $U(P) \leq U(0) = N(\hat{X}_0 - X|Y)$ where $p_{\hat{X}_0|Y} = p_{X|Y}$. Define $V \triangleq \hat{X}_0 - X$, then $\Sigma_{v|y} = \Sigma_{\hat{x}|y} + \Sigma_{x|y} = 2\Sigma_{x|y}$ where we use that $X$ and $\hat{X}$ are independent given $Y$. Thus,

$$U(0) = N(V|Y) \leq N(V_G|Y) = \left| \Sigma_{v|y} \right|^{1/d} = \left| 2\Sigma_{x|y} \right|^{1/d} = 2 \left| \Sigma_{x|y} \right|^{1/d} = 2N(X_G|Y), \tag{18}$$

where the first inequality is due to Lemma 1. Finally, for any $P \geq 0$ it holds that $U(P) \leq U(0)$ which implies $U(0) \leq 2N(X_G|Y)$, completing the proof.

## C  PROOF OF THEOREM 2

Assuming $D_v(X, \hat{X}|Y)$ is convex in its second argument, the constraint represent a compact, convex set. Moreover, $h(\hat{X} - X|Y)$ is strictly-concave w.r.t $p_{\hat{X}|Y}$ as a composition of a linear function (convolution) with a strictly-concave function (entropy). Therefore, we minimize a log-concave function over a convex domain and thus the global minimum is attained on the set boundary where $D_v(X, \hat{X}|Y) = P$.

## D  PROOF OF THEOREM 3

We begin with applying Lemma 1 and Lemma 2 to bound the objective function as follows

$$N(\hat{X}_g|Y) + N(X_g|Y) = N(\hat{X}_g - X_g|Y) \leq N(\hat{X} - X|Y) \leq N(\hat{X}_G - X_G|Y). \tag{19}$$

Note that the bounds are tight as the upper bound is attained when $\hat{X}|Y$ and $X|Y$ are multivariate Gaussian random variables, while the lower bound is attained if we further assume they are isotropic. Thus, we can bound the uncertainty-perception function as follows

$$U_g(P) \leq U(P) \leq U_G(P) \tag{20}$$

where we define

$$U_g(P) \triangleq \min_{p_{\hat{X}_g|Y}} \left\{ N(\hat{X}_g|Y) + N(X_g|Y) \ : \ D_{1/2}(X_g, \hat{X}_g|Y) \leq P \right\},$$

$$U_G(P) \triangleq \min_{p_{\hat{X}_G|Y}} \left\{ N(\hat{X}_G - X_G|Y) \ : \ D_{1/2}(X_G, \hat{X}_G|Y) \leq P \right\}. \tag{21}$$

The above quantities can be expressed in closed form. We start with minimization problem of the upper bound which can be written as

$$U_G(P) = \min_{p_{\hat{X}_G|Y}} \left\{ \frac{1}{2\pi e} e^{\frac{2}{d}\mathbb{E}[h(X_G - X_G|Y=y)]} \; : \; \mathbb{E}\left[D_{1/2}(X_G, \hat{X}_G|Y=y)\right] \leq P \right\}, \quad (22)$$

where the expectation is over $y \sim Y$. Substituting the expressions for $h(X_G - X_G|Y = y)$ and $D_{1/2}(X_G, \hat{X}_G|Y = y)$, we get

$$U_G(P) = \min_{\{\Sigma_{\hat{x}|y}\}} \left\{ \frac{1}{2\pi e} e^{\frac{2}{d}\mathbb{E}\left[\frac{1}{2}\log\left\{(2\pi e)^d |\Sigma_{\hat{x}|y} + \Sigma_{x|y}|\right\}\right]} \; : \; \mathbb{E}\left[\log \frac{\left|\left(\Sigma_{\hat{x}|y} + \Sigma_{x|y}\right)/2\right|}{\sqrt{|\Sigma_{\hat{x}|y}| \, |\Sigma_{x|y}|}}\right] \leq P \right\}. \tag{23}$$

Notice the optimization is with respect to the covariance matrices $\{\Sigma_{\hat{x}|y}\}$. Simplifying the above, we can equivalently solve the following minimization

$$\min_{\{\Sigma_{\hat{x}|y}\}} \mathbb{E}\left[\log \left|\Sigma_{\hat{x}|y} + \Sigma_{x|y}\right|\right] \text{ s.t. } \mathbb{E}\left[\log \frac{\left|\left(\Sigma_{\hat{x}|y} + \Sigma_{x|y}\right)/2\right|}{\sqrt{|\Sigma_{\hat{x}|y}| \, |\Sigma_{x|y}|}}\right] \leq P. \tag{24}$$

The solution of a constrained optimization problem can be found by minizmiation the Lagrangian

$$L\left(\{\Sigma_{\hat{x}|y}\}, \lambda\right) \triangleq \mathbb{E}\left[\log \left|\Sigma_{\hat{x}|y} + \Sigma_{x|y}\right|\right] + \lambda \left(\mathbb{E}\left[\log \frac{\left|\left(\Sigma_{\hat{x}|y} + \Sigma_{x|y}\right)/2\right|}{\sqrt{|\Sigma_{\hat{x}|y}| \, |\Sigma_{x|y}|}}\right] - P\right). \tag{25}$$

Since expectation is a linear operation and using that $P = \mathbb{E}[P]$, we rewrite the above as

$$L\left(\{\Sigma_{\hat{x}|y}\}, \lambda\right) = \mathbb{E}\left[\log \left|\Sigma_{\hat{x}|y} + \Sigma_{x|y}\right| + \lambda \left(\log \frac{\left|\left(\Sigma_{\hat{x}|y} + \Sigma_{x|y}\right)/2\right|}{\sqrt{|\Sigma_{\hat{x}|y}| \, |\Sigma_{x|y}|}} - P\right)\right]. \tag{26}$$

The expression within the expectation can be written as

$$\log \left|\Sigma_{\hat{x}|y} + \Sigma_{x|y}\right| + \lambda \left(\log \left|\left(\Sigma_{\hat{x}|y} + \Sigma_{x|y}\right)/2\right| - \frac{1}{2}\log \left|\Sigma_{\hat{x}|y}\right| - \frac{1}{2}\log \left|\Sigma_{x|y}\right| - P\right). \tag{27}$$

Next, according to KKT conditions the solutions should satisfy $\frac{\partial L}{\partial \Sigma_{\hat{x}|y}} = 0$. Using the linearity of the expectation and differentiating (27) w.r.t $\Sigma_{\hat{x}|y}$ we obtain

$$\left(\Sigma_{\hat{x}|y} + \Sigma_{x|y}\right)^{-1} + \lambda \left(\left(\Sigma_{\hat{x}|y} + \Sigma_{x|y}\right)^{-1} - \frac{1}{2}\Sigma_{\hat{x}|y}^{-1}\right) = 0 \tag{28}$$

Multiplying both sides by $\left(\Sigma_{\hat{x}|y} + \Sigma_{x|y}\right)$, we have

$$I + \lambda I - \frac{\lambda}{2}I - \frac{\lambda}{2}\Sigma_{x|y}\Sigma_{\hat{x}|y}^{-1} = 0$$

$$\Rightarrow (1 + \frac{\lambda}{2})I = \frac{\lambda}{2}\Sigma_{x|y}\Sigma_{\hat{x}|y}^{-1}$$

$$\Rightarrow (\lambda + 2)\Sigma_{\hat{x}|y} = \lambda\Sigma_{x|y} \tag{29}$$

$$\Rightarrow \Sigma_{\hat{x}|y} = \frac{\lambda}{\lambda + 2}\Sigma_{x|y}.$$

Define $\gamma = \frac{\lambda}{\lambda+2}$, so $\Sigma_{\hat{x}|y} = \gamma\Sigma_{x|y}$. Substituting the latter into the constraint we get

$$\log \left|\left(\gamma\Sigma_{x|y} + \Sigma_{x|y}\right)/2\right| - \frac{1}{2}\log \left|\gamma\Sigma_{x|y}\right| - \frac{1}{2}\log \left|\Sigma_{x|y}\right| = P$$

$$\Rightarrow n\log \frac{1+\gamma}{2} - \frac{n}{2}\log \gamma = P$$

$$\Rightarrow \frac{(1+\gamma)^2}{4\gamma} = e^{\frac{2}{d}P} \tag{30}$$

$$\Rightarrow \gamma^2 + 2\gamma + 1 = 4\gamma e^{\frac{2}{d}P}$$

$$\Rightarrow \gamma(P) = 2e^{\frac{2}{d}P} - 1 - \sqrt{(2e^{\frac{2}{d}P} - 1)^2 - 1}.$$

Thus, we obtain that

$$U_G(P) = \eta(P) \cdot N(X_G|Y) \tag{31}$$

where

$$\eta(P) = \gamma(P) + 1 = 2e^{\frac{2}{d}P} - \sqrt{(2e^{\frac{2}{d}P} - 1)^2 - 1}. \tag{32}$$

Notice that $\eta(0) = 2$, while $\lim_{P \to \infty} \eta(P) = 1$, so $1 \leq \eta(P) \leq 2$. Following similar steps where we replace $\Sigma_{\hat{x}|y}$ and $\Sigma_{x|y}$ with $N(\hat{X}|Y)$ and $N(X|Y)$ respectively, we derive

$$U_g(P) = \eta(P) \cdot N(X|Y). \tag{33}$$

## E    PROOF OF THEOREM 4

Define $\mathcal{E} \triangleq \hat{X} - X$. Then,

$$
\begin{aligned}
\frac{1}{d}\mathbb{E}\left[||\hat{X} - X||^2\right] &\underset{(a)}{=} \mathbb{E}\left[\frac{1}{d}\mathbb{E}\left[||\hat{X} - X||^2|Y\right]\right] = \mathbb{E}\left[\frac{1}{d}\mathbb{E}\left[||\mathcal{E}||^2|Y\right]\right] = \mathbb{E}\left[\frac{1}{d}\mathbb{E}\left[\mathcal{E}^T\mathcal{E}|Y\right]\right] \\
&= \mathbb{E}\left[\frac{1}{d}Tr\left(\mathbb{E}\left[\mathcal{E}\mathcal{E}^T|Y\right]\right)\right] = \mathbb{E}\left[\frac{1}{d}Tr\left(\Sigma_{\varepsilon|y}\right)\right] \\
&\underset{(b)}{\geq} \mathbb{E}\left[\left|\Sigma_{\varepsilon|y}\right|^{1/d}\right] = \mathbb{E}\left[\left|\Sigma_{\hat{x}|y} + \Sigma_{x|y}\right|^{1/d}\right] \\
&\underset{(c)}{\geq} \mathbb{E}\left[\frac{1}{2\pi e}e^{\frac{2}{d}h(\hat{X} - X|Y=y)}\right] \\
&\underset{(d)}{\geq} \frac{1}{2\pi e}e^{\frac{2}{d}\mathbb{E}[h(\hat{X} - X|Y=y)]} = \frac{1}{2\pi e}e^{\frac{2}{d}h(\hat{X} - X|Y)} = N\left(\hat{X} - X|Y\right),
\end{aligned}
$$

where (a) is by the law of total expectation, (b) is due to the inequality of arithmetic and geometric means, (c) follows Lemma 1, and (d) is according to Jensen's inequality.

