# OpenReview forum: "The Uncertainty-Perception Tradeoff"
_ICLR.cc/2024/Conference — Submitted to ICLR 2024_

### Official Review · Reviewer_dFBY · 2023-10-20

**Soundness:** 2 fair
**Presentation:** 3 good
**Contribution:** 2 fair
**Rating:** 6
**Confidence:** 3

**Summary:**

This paper introduces a new theoretical framework that captures the fundamental trade-off between perceptual reconstruction quality and estimator uncertainty in inverse problems. They show that as perceptual quality increases, the uncertainty of the estimator increases in tandem in the proposed information-theoretical formulation.They derive a feasible region in the trade-off plane that can be achieved by estimators, providing a potential venue to identify areas of improvement in existing reconstruction techniques. Some numerical experiments are provided on image data that demonstrates the trade-off.

**Strengths:**

- To the best of my knowledge, the proposed framework is novel in deriving the trade-off between perceptual quality and distortion metrics in inverse problems. The proxies to describe perceptual quality and estimator uncertainty are sensible. The direct tie to the perception-distortion trade-off is interesting.
- The paper is clearly written and fairly easy to follow.
- The investigated problem is crucial in better understanding the behavior and limitations of state-of-the-art image reconstruction methods that are able to produce exceedingly realistic images without providing clear ideas about the reliability of such results. Thus, the work is well-motivated and has a potential for improving our understanding of this fundamental trade-off.

**Weaknesses:**

- In my opinion, demonstrating how the framework can be used in practical settings is seriously lacking, somwehat undermining the potential impact and significance of the work. The authors claim that the introduced framework can be used to 1) assess estimator performance and 2) identify areas of improvement for image reconstruction techniques. Even though Figures 4 and 5 demonstrate that the framework gives sensible results that correspond to what we would expect from the theorems (higher perceptual quality results in higher uncertainty and higher distortion), it is not clear to me how much we can gain from this. In particular, I believe if we simply plotted LPIPS vs PSNR on various datasets for these techniques, we would see the same trend. That being said, the key contribution of the framework could be answering questions such as "How much can we improve the perceptual quality/distortion of this network based on our model of the trade-off?". Authors mention this direction repeatedly in the paper, but there are no experiments or further discussion supporting this. For instance, can we determine where a certain reconstruction technique lies in the uncertainty-perception plane and use that to derive useful clues what type of imrpovement we can still expect from our estimator? I believe that focusing on answering these questions on practical reconstruction techniques would very significantly increase the significance of the paper.
- The paper has multiple typos and there is a repeated sentence in Section 2, please check.

**Questions:**

- With respect to 1) in Weaknesses: Can the proposed framework reveal any insights about estimator performance that would not be possible by simply evaluating perceptual quality and distortion metrics?
- With respect to 2) in Weaknesses: How can the proposed framework be used to identify areas of improvement concretely in practice?

---

> ### Author Response · Authors · 2023-11-14
>
> We thank the reviewer for their detailed valuable feedback and constructive criticism, both of which we have carefully considered and addressed in our response below.
>
> Major Issues
> * Practical Implications:  We highly appreciate the reviewer's detailed feedback regarding the practical implications of our work. First, we completely agree with the reviewer that evaluating estimators’ performance using practical and established measures like LPIPS would yield similar trends to those presented. Our primary objective is to provide theoretical justification for such empirical observations by establishing a fundamental relationship between uncertainty and perception. This theoretical foundation serves as the core contribution of our research, with the numerical experiments primarily serving an illustrative purpose, aligning with previous works such as Blau & Michaeli (2018) and Freirich et al. (2021).
>
> Despite the theoretical focus, our work offers several practical implications -
>
> (1) Tradeoff Awareness: Developers should be aware of the tradeoff between high perceptual quality and low uncertainty when designing or optimizing restoration algorithms, prioritizing specific needs based on their application context.
>
> (2) Uncertainty-Perception Plane: Plotting algorithms on this plane, using any established measures of uncertainty and perception, should reveal potential areas for improvement. In our case, enhancing DIP's uncertainty without compromising perceptual quality presents a potential optimization opportunity. Additionally, attempts to improve EDSR's perception will likely increase its uncertainty, while lowering ESRGAN's uncertainty may compromise perceptual quality.
>
>
> Minor Issue - In response to the reviewer's feedback, we have carefully proofread the manuscript to ensure accuracy and clarity.
>
> The revised manuscript incorporates the above clarifications, addressing the reviewer’s concerns and refining our contributions. We hope these clarifications provide a clearer understanding of our work and improve its impact for the reviewer.

---

### Official Review · Reviewer_KFgG · 2023-10-28

**Soundness:** 3 good
**Presentation:** 3 good
**Contribution:** 2 fair
**Rating:** 5
**Confidence:** 2

**Summary:**

The paper investigates the tradeoff between perception and uncertainty in generative models used for restoration tasks. Specifically, authors employ information-theory tools to analyze this tradeoff and show that as the perceptual quality of a restoration algorithm increases, so does the uncertainty, as quantified by error entropy.

**Strengths:**

I would like to acknowledge that I am not familiar with the specific field to which this manuscript belongs. Therefore, the following assessment is based solely on my subjective perception of the paper.
The strength of this paper lies in its rigorous mathematical analysis of the tradeoff between perception and uncertainty in generative models for restoration tasks. By employing information-theory tools, the authors provide a principled analysis of uncertainty and its interplay with perception.

**Weaknesses:**

One potential weakness of this paper is the lack of empirical evaluation or experimental validation of the proposed analysis and tradeoff. I think the experiments in this article are not sufficient. Moreover, the results in Figure 4 do not effectively support the findings described in the text ‘methods that achieve low perceptual quality exhibit low uncertainty, while algorithms with superior perceptual quality result in high uncertainty values’, and the correlation between the two is not as strong as suggested.
To summarize, even though the paper provides a rigorous mathematical analysis of the tradeoff between perception and uncertainty, it seems that the paper lacks empirical evaluation or comparison with existing approaches, limiting the assessment of the practical implications of the analysis. Therefore, it feels like taking a few more steps of revision can be beneficial for the paper for improving its quality.

**Questions:**

N/A

---

> ### Author Response · Authors · 2023-11-14
>
> We appreciate the reviewer for taking the time to review our manuscript, and for their honesty in acknowledging their limited experience in this field. Even from a subjective viewpoint, these perspectives are helpful for broadening the reach and clarity of our research.
>
> Major Issues
>
> * Empirical Results: We appreciate your feedback regarding the empirical aspects of our work. We want to emphasize that our primary contribution lies in the theoretical domain, focusing on analyzing the relationship between uncertainty and perception in restoration tasks. Despite this theoretical focus, our findings offer valuable practical implications -
>
> (1) Tradeoff Between Perception and Uncertainty: Developers should consider this tradeoff when designing or optimizing restoration algorithms, prioritizing either high perceptual quality or low uncertainty based on specific needs.
>
> (2) Improvement Opportunities: Figure 4 showcases potential areas for improvement in specific algorithms. For instance, we might be able to enhance DIP's uncertainty without compromising perceptual quality. Additionally, attempts to improve EDSR's perception will likely increase its uncertainty, while lowering ESRGAN's uncertainty will likely decrease its perceptual quality.
> Following previous works by Blau & Michaeli (2018) and Freirich et al. (2021), we utilize super-resolution algorithms as an illustrative example to demonstrate our theoretical framework. While we acknowledge the value of additional experiments, such extensive empirical investigations are beyond the scope of this paper, which prioritizes theoretical contributions. We envision future research, similar to "The 2018 PIRM Challenge on Perceptual Image Super-resolution," to delve deeper into empirical experiments.
>
>
> * Figure 4: We agree that the text in the current manuscript does not accurately reflect the findings in Figure 4. We apologize for the inaccurate wording and will revise it in the next version. We would like to clarify the following points -
> The blank region in the lower left corner in Figure 4 represents an unattainable region in the uncertainty-perception plane. This supports the claim that no model can simultaneously achieve both low uncertainty and high perceptual quality.
> In the vicinity of this blank region, we observe an anti-correlation between uncertainty and perception. As perceptual quality modestly improves, uncertainty dramatically increases, suggesting a tradeoff between these two aspects.
>
> These clarifications are incorporated into the revised manuscript, aiming to address the reviewer's concerns and further strengthen our contributions. We hope they improve the understandability and impact of our work for the reviewer.

---

### Official Review · Reviewer_afrr · 2023-10-30

**Soundness:** 3 good
**Presentation:** 2 fair
**Contribution:** 2 fair
**Rating:** 3
**Confidence:** 4

**Summary:**

This paper presents a rigorous theoretical analysis of the tradeoff between uncertainty and perceptual quality in generative models for ill-posed inverse problems like image restoration. Leveraging information theory tools, the authors introduce an uncertainty-perception (UP) function that captures the minimal uncertainty for a given level of perceptual quality. They prove several valuable properties of this function, establishing its quasi-linearity and bounding behavior. By adopting Renyi divergence as the perceptual measure, they further derive analytical bounds confining the UP function, giving rise to an insightful uncertainty-perception plane. This geometric construction categorizes estimators into impossible, optimal and suboptimal regions. The analysis reveals a dependence on dimensionality, with the tradeoff becoming more severe for higher dimensions. Finally, the authors connect uncertainty to MSE distortion, offering a novel perspective on the classic distortion-perception tradeoff. Experiments on super-resolution methods validate the tradeoff in practice.

**Strengths:**

•	Provides novel theoretical framework to analyze uncertainty-perception tradeoff based on information theory principles.
•	Establishes and proves existence of inevitable tradeoff through rigorous analysis.
•	Derives insightful analytic bounds confining UP function to convex envelopes.
•	Uncertainty-perception plane offers intuitive visualization and practical utility for assessing estimators.
•	Connects uncertainty to distortion, offering new view on classic distortion-perception tradeoff.

**Weaknesses:**

•	Assumptions like unbiasedness and Markov chain may limit applicability in some cases.
•	More analysis for other divergence measures besides Renyi could strengthen claims.
•	Additional validation on diverse restoration tasks needed to fully support general claims.

**Questions:**

•	Does the proposed strategy work only for image super-resolution? Could it work for other models and tasks?
•	Can the theory guide development of new algorithms to achieve better uncertainty-perception tradeoffs?

---

> ### Author Response · Authors · 2023-11-14
>
> We thank the reviewer for reviewing our manuscript and providing valuable feedback and constructive criticism. We have carefully considered all the reviewer's concerns and addressed each point in the following response.
>
> Major Issues
>
> * Assumptions:
> (1) Unbiasedness - this assumption holds in our case without loss of generality, since our uncertainty measure, power entropy, is translation-invariant and thus insensitive to bias.
> (2) Markov Chain - this assumption suggests that the estimation process effectively captures all relevant information about X from Y. As we do not have access to the true signal X, this is a reasonable assumption which aligns with  existing literature on restoration tasks, in particular with [Blau & Michaeli 18] and [Freirich et al. 21]). Intuitively, the details in the estimation that are independent of the true image can be seen as "hallucinations" that contribute to improved perceptual quality. However, these same details also lead to higher uncertainty, as they deviate from the ground truth and introduce additional variability into the estimation.
> We assure the reviewer that the revised manuscript includes an extended discussion of these assumptions and clarifies the points mentioned above to ensure better understanding and transparency.
>
> * Additional Divergence Measures: We appreciate the reviewer's suggestion to explore other divergence measures. Following the approach of Blau & Michaeli (2018), we first define the uncertainty-perception function using a theoretic definition of perceptual quality as a divergence between the true and estimated distributions. This allows us to establish general properties of the function. Then, we follow the example of Freirich et al. (2021) and consider a specific perception measure, Renyi divergence. This choice aligns with widely used measures like KL divergence and Hellinger distance, enabling us to derive further insights about the uncertainty-perception function. We believe this approach ensures consistency with existing literature and applicability to many common cases. We emphasize these points in the revised manuscript.
>
> * Theory to Practice: First, we would like to emphasize that the primary focus of our work is theoretical analysis and its contribution lies in illuminating the relationship between uncertainty and perception in restoration tasks. Nevertheless, our theoretical results can offer practical insights for algorithms, both existing and new. The tradeoff we identified suggests that high perceptual quality usually leads to higher uncertainty, and vice versa. Developers should consider this tradeoff when setting requirements for their solutions. For example, examining Figure 4 suggests opportunities for improving DIP in terms of uncertainty without sacrificing perceptual quality. Additionally, attempts to improve EDSR's perception will likely increase its uncertainty, while lowering ESRGAN's uncertainty will likely come at the cost of decreased perception.
> We address these points in the revised manuscript, where we broaden the relevant discussions and emphasize the theoretical focus of our work. Furthermore, we will revise the title to " A Theory of Uncertainty-Perception Tradeoff" of the camera-ready version to better reflect this emphasis.
>
> * Generalization to Other Models and Tasks:  We reiterate that the primary focus of this paper is the theoretical examination of the interplay between uncertainty and perception. Following Blau & Michaeli (2018) and Freirich et al. (2021), we utilize super-resolution algorithms as an illustrative example to demonstrate our theoretical results. Our analysis does not rely on any specific task or model and thus theoretically generalizes to other settings. We agree that further experiments involving various tasks and models would be valuable. However, as our current work emphasizes theoretical contributions, we leave such extensive empirical investigations for a future paper, similar to "The 2018 PIRM Challenge on Perceptual Image Super-resolution."
>
> We hope that these clarifications address the reviewer's concerns comprehensively and clarify the theoretical impact and applicability of our work.

---

### Official Review · Reviewer_WDqn · 2023-10-31

**Soundness:** 3 good
**Presentation:** 4 excellent
**Contribution:** 3 good
**Rating:** 8
**Confidence:** 4

**Summary:**

This work presents a generalization of the distortion-perception tradeoff. The (previously reported) distortion-perception tradeoff itself is an extension of the classical distortion-rate curve (or tradeoff) in signal reconstruction by considering the preservation of the signal PDF. It turns out that (surprisingly) better preservation of the PDF raises the distortion-rate curve. As perceptual quality is somehow related to PDF preservation, divergence between the PDFs of the original and reconstructed signal is called "perceptual deviation". The distortion-"perception" tradeoff is actually a distortion-divergence tradeoff.

In this work, the authors extend the distortion-perception tradeoff by changing the distortion term. As opposed to the expected value of a distance between original and reconstructed signals, the authors propose to measure distortion through the entropy of the reconstruction error. They propose to measure distortion through uncertainty (or variability) as opposed to distance. In this way they define the uncertainty function U(P), that depends on P (Perception = divergence between PDFs); as opposed to the (previosuly used) distortion function D(P). The authors prove the properties of the U(P) function and show that D(P)>= U(P).

**Strengths:**

* Well written: a pleasure to read!.

* Extends a previous interesting concept, and the new results are consistent with previous reports. Effects of signal estimation on (1) the deviation of the estimate from the original sample and on (2) the deviation of the estimated PDF from the original PDF, are interesting points for the ICLR community (because of its implications in inference, generative models...), and this work further digs into these issues.

* The properties of the new tradeoff are analytically proved, so results are solid. While the main concept (the joint consideration of both deviations 1 and 2) was introduced in a previous work, it is interesting to keep on pointing out the surprising relation between the two deviations and this (technically sound) work presents analytical properties of an information theoretic extension of the distortion concept in the distortion-perception framework.

**Weaknesses:**

* Introduction is confusing for those not familiar with the "perceptual quality" concept (divergence between PDFs) introduced in [Blau & Michaeli 18].

* Perceptual distances between samples (the regular perceptual quality concept) have been related to the signal PDFs as well.

* Experimental illustration of super resolution is limited: no visual examples are given. Note that the "perceptual quality" concept defined from divergence is an abstraction. Actual assessment of the perceptual quality should be done through visual inspection, but no visual examples are given.

* Numerical evaluation is prone to error because entropy and divergence values are based on estimations from samples in high-dimensional scenarios. This should be acknowledged.

**Questions:**

MAJOR ISSUES

* The introduction should include a citation to the "perceptual quality" concept defined from divergence between PDFs as done in [Blau & Michaeli 18, 19] because if not, the key question "Can one design an AI model of high perceptual quality which exhibits low uncertainty" is unclear or vague.

* In the same vein, the "related work" section should include a paragraph on "Perceptual quality quantification". This section should link the Blau&Michaeli definition with "no-reference distance" definitions based on the similarity between PDFs (e.g. citations to [32-34] and associated reasoning in [Blau & Michaeli 18]).
However, note that this divergence concept is related to the usefulness of summary statistics to capture the nature of textures, as in [Portilla&Simoncelli IJCV 2000], in more recent style transfer algorithms based on difference of Gram matrices in VGG-like nets [Gatys et al. CVPR 2016]) or in state-of-the-art perceptual distortion metrics such as DISTS [Ding et al. IEEE PAMI 2020]. Also worth citing in that paragraph is the recent work relating perceptual distances and PDFs [Hepburn et al. ICLR 2022], which makes interesting points on the difference between individual distances between samples and averages over ensembles. In particular [Hepburn et al. ICLR 2022] shows that perceptual distances capture relevant information on the image PDF.

* Please discuss if the proposed uncertainty-perception concept could be extended to rate-distortion?  R(U,P) similarly to R(D,P) in [Blau & Michaeli 19].

* The experiments depend on estimations of entropy and divergence between PDFs from samples of 243 dimensions. These estimations are risky and prone to high bias. How many samples did you used? Have you checked other estimators appart from Kozachenko-Leonenko?. Ready-to-use alternatives include (1) an improved Kozachenko-Leonenko estimator [Marin-Franch & Foster IEEE PAMI 2013] available here https://github.com/imarinfr/klo , or (2) a Gaussianization-based algorithm which has proved to be better [Laparra et al. IEEE TNN 2011], see the comparisons in [Malo J. Math. Neurosci. 2020], or in [Laparra et al. 2023 https://arxiv.org/abs/2010.03807], available here https://isp.uv.es/RBIG4IT.htm    https://github.com/IPL-UV/rbig
I suggest to repeat Fig. 4 and 5 with other estimators to have stronger evidences of the trend.

* Please include a visual example for several points of Fig. 4. At least in the appendix if it doesnt fit in the main text. Remember that beyond conceptual definitions of "Perceptual quality", in this regard, nothing substitutes visual inspection of representative examples.
In these examples please report the uncertainty (entropy of error) but also the MSE.

MINOR ISSUES

* Why in fig 5 we have less points than in fig. 4?

* The third paragraph in page 2 has a repeated sentence "Conformal methods..."

* Last sentence of the fourth paragraph of page 2 is confusing (particularly as "perceptual quality" as in Blau&Michaeli had not been defined). Currently it says "While the above studies address both uncertainty and perception, none of them explicitly quantify
uncertainty as a function of perceptual quality"... Please clarify in which way the above studies talk about "perception" and how your work is different. Is it because this one uses the divergence definition?. Probably it will be easier to clarify this if the suggested paragraph on "Perceptual Quality Quantification" is added before this comment.

* What is n in the exponent of Lemma 1? is it dimension d?

* [Blau & Michaeli 18a] and [Blau & Michaeli 18b] are the same?

---

> ### Author Response · Authors · 2023-11-14
>
> We are grateful to the reviewer for the insightful comments and constructive criticism. The detailed feedback has been incredibly helpful in revising and strengthening our manuscript. We have carefully considered all the concerns raised and addressed them point-by-point below.
>
> Major Issues
>
> * Perceptual Quality: We appreciate the reviewer's concern about the potential ambiguity of "perceptual quality" and the suggestion to clarify its definition in our manuscript. We agree that quantifying this concept is a complex and evolving field with diverse approaches. To ensure consistency with previous works, we follow the theoretical definition of perceptual quality as the divergence between probability density functions (PDFs), as proposed by [Blau & Michaeli 18, 19]. This approach aligns with relevant prior work and establishes a clear framework for our analysis.
> Following the reviewer's feedback, we have clarified the above in the Introduction explicitly stating our chosen definition of perceptual quality and its connection to PDF divergence. We have added a new paragraph on "Perception Quantification" discussing the various ongoing efforts and diverse approaches in this field.
>
> * Rate-Distortion: We strongly believe the uncertainty-perception tradeoff can be extended to incorporate rate, given the close connection between mutual information and entropy. While we do not have concrete theoretical results currently, we briefly mention this potential future direction in the revised manuscript, as suggested by the reviewer.
>
> * Estimation in High Dimension: We agree with the reviewer's concern about the challenges of entropy and divergence estimation in high dimensions. Following previous works by [Blau & Michaeli 18] and [Freirich et al. 21], we perform our estimations at the patch level, where we have more than 16 million samples. However, we acknowledge that challenges and potential limitations remain associated with high-dimensional estimations, which we explicitly state in the revised manuscript.  We highly appreciate the reviewer's suggestions and pointers to improved estimators, which we hope to include in the camera-ready version.
>
> * Visual Examples: We agree with the reviewer's comments and accept their suggestion to include a visual comparison of various estimators concerning uncertainty and perception in the main body of the paper.
>
> Minor Issues
>
> * We have updated both figures to have the same number of data points.
> * We have removed repeated sentences.
> * We have clarified the manuscript to explicitly state that the discussed studies do not analyze perceptual quality or its relation to uncertainty.
> * We have fixed the duplicate citation error.
>
> We hope the above addresses the reviewer's concerns effectively and improves the clarity, accuracy, and overall quality of our manuscript.

---

### Author Response · Authors · 2023-11-23
**Main Revisions Following Rebuttal**

We deeply appreciate the reviewers' insightful feedback, which significantly helped us refine and strengthen our manuscript. Below, we summarize the main revisions made in response to your valuable comments:

1. The manuscript has been revised to emphasize the theoretical focus of the work.
2. Nonetheless, the theoretical findings offers practical implications as developed should consider the tradeoff when designing restoration algorithms, prioritizing either high perceptual quality or low uncertainty.
3. Visual examples comparing various estimators concerning uncertainty and perception are added.
4. Figures are updated with consistent data points, and experiments are extended with recent algorithms.
5. The theoretical definition of "Perceptual quality" is clarified, and existing approaches to quantify it are discussed.
6. We have extended the discussions about various topics: underlying assumptions, generalization to other inverse problems, estimating statistics in high-dimension, etc.

Further revisions and detailed responses are given per-reviewer below.

We sincerely hope these revisions address your concerns and effectively respond to the valuable feedback you provided. We believe these changes significantly improve the clarity, accuracy, and overall impact of our work.

---

### Meta-Review · Area_Chair_RVih · 2023-12-08

**Metareview:**

Although the reviewers raise a number of critical points in their original reports, there is agreement that the paper makes an interesting and potentially useful contribution. The theoretical analysis of the tradeoff between perception, distortion, and model uncertainty is undoubtedly valuable. The authors showed a high level of commitment during the rebuttal phase and did their best to respond to the comments and to improve the submission. This was appreciated and positively acknowledged by all. In the discussion between authors and reviewers, some critical points could be resolved and some questions clarified. Other points remained open and were critically reconsidered in the subsequent internal discussion.

Eventually, there was a consensus that the paper still remains a bit behind the expectations for a top-venue such as ICLR. Although the theoretical focus is acknowledged, it would also important to highlight the practical implications and advancements of the proposed technique. There is also a lack of sufficient evaluations and comparisons with existing methods.

**Justification For Why Not Higher Score:**

Practical implications and advancements of the proposed technique remain unclear. There is also a lack of sufficient evaluations and comparisons with existing methods.

**Justification For Why Not Lower Score:**

N/A

---

### Decision · Program_Chairs · 2024-01-16

Reject